# Protective Effect and Mechanism of Xbp1s Regulating HBP/O-GlcNAcylation through GFAT1 on Brain Injury after SAH

**DOI:** 10.3390/biomedicines11051259

**Published:** 2023-04-24

**Authors:** Kefan Wu, Lili Chen, Zhen Qiu, Bo Zhao, Jiabao Hou, Shaoqin Lei, Meng Jiang, Zhongyuan Xia

**Affiliations:** Department of Anesthesiology, Renmin Hospital of Wuhan University, Wuhan 430064, China; wkf7788@whu.edu.cn (K.W.);

**Keywords:** subarachnoid hemorrhage, Xbp1s, O-GlcNAc, HBP

## Abstract

(1) SAH induces cellular stress and endoplasmic reticulum stress, activating the unfolded protein response (UPR) in nerve cells. IRE1 (inositol-requiring enzyme 1) is a protein that plays a critical role in cellular stress response. Its final product, Xbp1s, is essential for adapting to changes in the external environment. This process helps maintain proper cellular function in response to various stressors. O-GlcNAcylation, a means of protein modification, has been found to be involved in SAH pathophysiology. SAH can increase the acute O-GlcNAcylation level of nerve cells, which enhances the stress capacity of nerve cells. The GFAT1 enzyme regulates the level of O-GlcNAc modification in cells, which could be a potential target for neuroprotection in SAH. Investigating the IRE1/XBP1s/GFAT1 axis could offer a promising avenue for future research. (2) Methods: SAH was induced using a suture to perforate an artery in mice. HT22 cells with Xbp1 loss- and gain-of-function in neurons were generated. Thiamet-G was used to increase O-GlcNAcylation; (3) Results: Severe neuroinflammation caused by subarachnoid hemorrhage leads to extensive endoplasmic reticulum stress of nerve cells. Xbp1s, the final product of unfolded proteins induced by endoplasmic reticulum stress, can induce the expression of the hexosamine pathway rate limiting enzyme GFAT1, increase the level of O-GlcNAc modification of cells, and have a protective effect on neural cells; (4) Conclusions: The correlation between Xbp1s displayed by immunohistochemistry and O-GlcNAc modification suggests that the IRE1/XBP1 branch of unfolded protein reaction plays a key role in subarachnoid hemorrhage. IRE1/XBP1 branch is a new idea to regulate protein glycosylation modification, and provides a promising strategy for clinical perioperative prevention and treatment of subarachnoid hemorrhage.

## 1. Introduction

Subarachnoid hemorrhage (SAH) is the third most common subtype of stroke. SAH accounts for only 5% of strokes but occurs at a fairly young age [1]. Although the incidence rate of subarachnoid hemorrhage has gradually decreased in recent years, the current research has been focused on more effective clinical protective measures. Neuroinflammation and oxidative stress play important roles in early brain injury following subarachnoid hemorrhage (SAH) [2,3]. At the same time, subarachnoid hemorrhage will lead to endoplasmic reticulum stress of nerve cells [4]. Persistent endoplasmic reticulum stress will induce unfolded protein response (UPR). UPR mainly includes three pathways—ATF6 pathway, IRE1 pathway, and PERK pathway. ATF6 pathway and Xbp1 pathway express an endoplasmic reticulum stress partner and strengthen the ability to fold proteins. If the endoplasmic reticulum stress state continues, the final product of the PERK pathway will induce the global inhibition of cell translation [5,6,7].

Regulatory protein modification is a new idea for clinical treatment of subarachnoid hemorrhage. By regulating the protein modification of key proteins, the protein activity can be increased, and the related pathway proteins can be activated to improve the cell stress ability. Among the final products of the unfolded protein reaction, XBP1s couples the UPR with the hexosamine pathway (HBP) by regulating expression of major HBP enzymes-GFAT1; HBP produces uridine diphosphate N-acetylglucosamine (Udp-GlcNAc), which is the substrate for O-GlcNAc modification (O-GlcNAcylation) [8,9]. In summary, the role of the XBP1s/HBP/O-GlcNAc axis in the nervous system has been clarified. However, whether it has potential therapeutic value in subarachnoid hemorrhage and can be regulated by other proteins deserves further study. The purpose of this experiment is to verify the role of the final product of the unfolded protein reaction induced by subarachnoid hemorrhage in improving the level of glycosylation of nerve cells and explore its neuroprotective mechanism.

## 2. Materials and Methods

### 2.1. Animals

All animal study protocols were approved by the Animal Ethics Committee of Wuhan University of China (WDRM20220802A, 20 August 2022 to 19 August 2023). C57BL/6 mice were purchased from Wuhan Chunyuhong Experimental Animal Feed Company. Mice weighing 25–30 g were maintained at 25 °C and 50% humidity. The rats had free access to food and water and lived in individual ventilated cages under specific-pathogen-free conditions in the Animal Facility of the Experimental Research Center of Wuhan University of China (license No. SCXK(Jing) 2014-0004).

### 2.2. SAH Model

All animals received adaptive feeding for several days. We used a random number table to allocate participants into different groups. After anesthesia with 50 mg/kg of pentobarbital sodium, mice were placed in a heat blanket to maintain the rectal temperature at 37 ± 0.5 °C. The internal carotid artery, external carotid artery, and left common carotid artery were then exposed through a middle-of-the-neck incision. The left external carotid artery was ligated and cut to leave a 3 mm arterial stump. A nylon suture was inserted into the left carotid through the middle cerebral artery branch, and the suture was further advanced by approximately 3–5 mm to perforate the artery. The head of the nylon suture was shaped via scalpel to perforate the artery more easily. The filament was withdrawn after 10 s to induce bleeding. Thermometer and heat preservation lamp were used during the operation to strictly control the temperature of the mice, and then placed in a 37 °C incubator for reperfusion after the operation. The sham operation was only performed for opening the neck. Thiamet G (500 mg/kg/d, Adooq, Irvine, CA, USA) was given by intraperitoneal injection 1 h prior to the operation. The animals were returned to their cages following the operation for recovery.

### 2.3. Weighing of Brain Edema

Animals were sacrificed after 24 h following induction of SAH and the brains were quickly removed with the right and left cerebral and cerebellar hemispheres isolated. The separate parts of the brain were weighed (wet weight), placed inside the oven at 100 °C for 24 h, and then re-weighed (dry weight). Water content can be calculated according to the formula: (wet weight − dry weight)/wet weight × 100%.

### 2.4. HE Staining

Paraffin-embedded mouse brain tissue sections were dewaxed in xylene and rehydrated in graded alcohols. The sections were then stained with hematoxylin solution for 5–10 min, followed by rinsing in tap water. The sections were subsequently dipped in 1% acid alcohol, rinsed again in tap water, and stained with eosin solution for 2–5 min. The sections were then dehydrated in graded alcohols and cleared in xylene before being mounted with mounting medium and coverslips. All steps were performed in a well-ventilated area with appropriate personal protective equipment. The staining protocol was optimized for mouse brain tissue and the staining times were selected to achieve the desired staining intensity.

### 2.5. Construction and Staining of Immunochip

Fresh tissue was fixed with formaldehyde for more than 24 h. The fixed tissue was trimmed in the fume hood prior to being placed into the dehydration box (DIAPATH, type: Donatello, Italian). The dehydration box was dehydrated successively with gradient alcohol. The waxed tissue was embedded in the embedding machine. After being cooled at −20 °C on a freezing table, the wax block was removed from the embedding frame and trimmed after the wax was solidified. (75% alcohol for 4 h; 85% alcohol for 2 h; 90% alcohol for 2 h; 95% alcohol for 1 h; anhydrous alcohol I for 30 min; anhydrous alcohol II for 30 min; alcohol benzene for 10 min; xylene I for 10 min; xylene II for 10 min; 65 °C melted paraffin I for 1 h; 65 °C melted paraffin II for 1 h; and 65 °C melted paraffin III for 1 h).

A sampler was used to sample the CA1, DG, and EC regions of the mouse brain to obtain the tissue columns. The obtained tissue columns were placed in rows according to the control group, SAH group, and the SAH+Thiamet-G group, and injected into the corresponding wax block holes in sequence. The tissue chip fusion instrument (Servicebio, type: JX-10, Wuhan, China) was used to repeatedly fuse the tissue column point and the recipient wax block so that the two were completely fused into the tissue chip wax block. The trimmed tissue chip wax block was placed on the paraffin sectioner for sectioning, with the thickness of 4 μm. The slices were floated in the 40 °C warm water of the spreader to flatten the tissue, and the tissue was added to a slide and placed in a 60 °C oven to bake.

After cooling sections to room temperature, the slides were washed by PBS (pH 6.5). After being blocked in normal goat serum for 30 min, the slides were incubated overnight at 4 °C with polyclonal antibodies. (Anti-c-Fos rabbit polyclonal antibody, #GB11069, 1:200, Servicebio; Anti-GFAP rabbit polyclonal antibody, #GB11096, 1:200, Service bio; Xbp1s rabbit monoclonal antibody, #40435S, 1:400, cell signaling technology; ATF-6 rabbit monoclonal antibody, #65880S, 1:200, cell signaling technology; O-GlcNAc rabbit monoclonal antibody, #82332S, 1:400, cell signaling technology). After washing the slides with PBS, they were incubated for 120 min at room temperature with goat anti-rabbit IgG (HRP-conjugated goat anti-rabbit IgG; Cy3 conjugated goat anti-rabbit IgG) and washed with PBS. Anti-fluorescence quenching sealing agent (containing DAPI) was placed on each slide, and all the slides were observed and analyzed under a BX51 microscope (5×, 20×, 40×, Olympus, Tokyo, Japan).

### 2.6. Open Field

An experimental box (50 cm × 50 cm × 40 cm) was cleaned prior to the experiment. The recovered mice were permitted to adapt to the box environment for 3 h. All the parameters were set in the Anymaze software. The weak light was used during the experiment. After the adaptation, the mice were placed into the central area of the experimental box and began recording mouse activity for 5 min. Horizontal or vertical activity was significantly lower than the average level if the same batch of animals had been abandoned. After the test, Anymaze was used to analyze the data.

### 2.7. HT22 Cell Culture and Transfection

The standard mouse hippocampal neuron cell line (HT22, source: Wuhan University Cell Library) was recovered on the 6th floor of the central laboratory of the People’s Hospital of Wuhan University. After the cells were stable, the cells were recovered by 5 × 10^4^/mL density on a culture plate (96-well plate, 100 μL/hole; 6-well plate, 2 mL/well) or petri dish (6 cm). The cells were divided into groups by random number method.

The cells were cultured in a non-glucose medium and exposed to a gas mixture containing 94% N2, 5% CO_2_, and 1% O_2_ for 6 h to simulate oxygen-glucose deprivation. Following this, they were returned to normoxic conditions (normal air) for a 24-h re-oxygenation period. The HT22 OGD model is widely accepted in neuroscience research for studying cellular and molecular mechanisms in ischemic brain injury. By using this model, we can investigate SAH pathophysiology and potential therapeutic targets, ultimately contributing to a better understanding of the disease and the development of novel treatments.

We utilized sgRNA to silence the expression of Xbp1 in Ht22 cells while concurrently employing a plasmid to overexpress Xbp1. A control group with an empty vector was also included for comparison. The cells were evenly planted in the well plate with appropriate density. Before transfection, the original cell culture medium was removed. After 1-2 times of PBS cleaning, 1.8 mL of basic culture medium without serum was replaced. Ctrl-SgRNA groups were inoculated with pLenti-Control-sgRNA. Xbp1-SgRNA groups were inoculated with lentivirus pLenti-CRISPRv2-sgRNA targeting Xbp1 [10,11]. Ctrl-plasmid groups were inoculated with 8 μL PEI 40 K transfection reagent, Xbp1-plasmid were inoculated 2 μg plasmid DNA (xbp1-overexpressed plasmmid, pHBLPm002838, Hanheng, China), and 8 μL PEI 40 K were inoculated with transfection reagent.

### 2.8. Flow Cytometry

Cells in the orifice plate were digested, collected in a flow tube, and washed with PBS three times. The cells were then centrifugated at 1500 rpm for 5 min. The cells were resuspended by 1× binding buffer. 5 μL Annexin V-FITC and 5 μL Propidium Iodide were added into the flow tube and reacted in the dark for 15 min at room temperature. After dyeing and incubation, 400 μL 1 × BindingBuffer working solution was added to each tube. After mixing well, the cells were detected the green fluorescence of Annexin V-FITC through FITC channel (FL1), and PI red fluorescence through PI channel (FL2). The parameters of flow cytometry (CytoFLEX, Beckman Coulter, Brea, California, USA) were as follows: excitation wavelength Ex = 488 nm, emission wavelength FL1 (Em = 525 ± 20 nm), and FL2(Em = 585 ± 21 nm). The proportion and number of cells in different states (mainly early withering, late withering, and death) were analyzed with FlowJo software.

### 2.9. Cells Immunofluorescence Staining

The cells climbed onto coverslip in the 6-well plate. After OGD treatment, the culture medium was removed, washed twice with PBS, and fixed with 1 mL of formaldehyde for 20 min. Then, the slides were washed with PBS and added 0.1% Triton 1 mL to break the membrane. Following the PBS wash, 0.5 mL 5% BSA was added for 1 h, and the rat-anti-O-GlcNAc antibody (1:100) was added at 4 °C overnight. Following an additional PBS wash for three times, fluorescent secondary antibody (1:200) was added for incubation in the dark at room temperature for 1 h, and photos were taken under a microscope. (400×, Olympus, Japan).

### 2.10. Western Blot Assay

After the completion of cell reoxygenation, the cells was washed by PBS to remove the residual culture medium; then, Rippa lysate was added, cells from the culture dish were scraped, ultrasonic-cracked on ice for 1 min (7 s, 3 s, 37%), centrifuged at low temperature, and boiled in the supernatant with sample buffer for 10 min. After PAGE gel electrophoresis, the proteins were transferred to PVDF membrane. The PVDF membrane was wobbled in 5% skimmed milk powder. After undergoing a PBS wash, the PVDF membrane was cut into bands according to the molecular size of the target protein. The bands were put into EP tubes containing Xbp1\GFAT1\GAPDH (Xbp1s rabbit monoclonal antibody, 1:1000, cell signaling technology; Anti-GFAT1 rabbit polyclonal antibody, 1:1000, Servicebio; Anti-GAPDH rabbit polyclonal antibody, 1:2000, Servicebio) overnight. The bands were washed three times and incubated at room temperature for 1 h with fluorescent secondary antibody (1:10000). The fluorescent protein bands were analyzed by Odessy(LI-COR Biosciences, USA), and the gray value of each target protein band was determined. The expression level of the target protein was expressed by the ratio of the gray value of the target protein band to the gray value of the GAPDH band.

Two hours after the mice had fully recovered from anesthesia, brain tissue was washed with precooled PBS and stored in liquid nitrogen. 100 mg of brain tissue was placed in RIPA lysate for decomposition and homogenized with ultrasonic lysate. The mixed liquid was put on ice for 20 min, and the supernatant was boiled with sample buffer for 10 min. The next step was similar to cell protein. The detected antibodies included P-perk (Anti-P-perk rabbit monoclonal antibody, #3179S, 1:1000, cell signaling technology, China), Chop (Anti-chop rabbit monoclonal antibody, #5554S, 1:1000, cell signaling technology, China), Bip (rabbit polyclonal to GRP78 BiP, #ab21685, 1:1000, Abcam, USA), Xbp1s (Xbp1s rabbit monoclonal antibody, #40435S, 1:1000, cell signaling technology, China), ATF6 (ATF-6 rabbit monoclonal antibody, #65880S, 1:1000, cell signaling technology, China).

### 2.11. Statistical Analysis

All data analyses were performed with Prism 9.5 (GraphPad Software, version 9.5.0-730). Data are presented as mean ± SEM. One-way ANOVA and Fisher’s LSD test, in that order, were applied for multiple comparisons, and the Student’s t-test was utilized for comparison between two groups. The individual data of brain edema weight and total distance are also shown in the figures. The level of significance was set at *p* < 0.05.

## 3. Results

### 3.1. Endoplasmic Reticulum Stress Induced by SAH Caused Serious Nerve Damage and Behavioral Changes, Which Could Be Alleviated by Thiamet-G

Brain tissue that received SAH operation showed obvious blood in the subarachnoid space. HE staining of brain tissue showed that there were red blood cells accumulated in the subarachnoid space of SAH group, indicating that the SAH model had achieved success (Figure 1a,b). In the brain edema data, there was a difference between the control group and the SAH group (*p* < 0.01), and there was a difference between the SAH group and the SAH+Thiamet-G group (*p* < 0.01). The water content of the brain tissue in the SAH group was significantly higher than that in the control group, indicating that the degree of edema was higher than the control group, while the brain tissue edema was reduced by the pretreatment of Thiamet-G (Figure 1c). In the open field experiment, there was a difference between the control group and the SAH group, and there was a difference between the SAH group and the SAH+Thiamet-G group (*p* < 0.01). The mice in the control group had regular activities and large activity tracks. The activities of the mice undergoing the SAH operation were disordered, the activity tracks were significantly reduced, and they were easily confined to a certain part in a short time. The use of Thiamet-G increased the activity tracks of the mice (Figure 1d). In the western blot assay, the expression levels of P-Perk, Bip, Chop, Xbp1s, and ATF6 in the SAH group were higher than those in the control group. (*p* < 0.01), while the expression levels of p-Perk, Bip, and Chop in the SAH+Thiamet-G group were lower than those in the SAH group (*p* < 0.01). It was noteworthy that the difference between Xbp1s and ATF6 was not significant (*p* > 0.01) (Figure 2a,b).

### 3.2. Immunofluorescence Staining of Tissue-Chip Showed the Expression and Distribution of c-fos, GFAP, ATF6, Xbp1, and O-GlcNAc Modification in Different Brain Regions

Dapi staining was used for each immunostaining to mark the contour of brain and cell distribution. The immunofluorescence intensity is expressed by the ratio of the immunofluorescence intensity of the target region to the CA1 region of the control group (Figure 2e, Figure 3c, Figure 4c and Figure 5c,d). The activated state of astrocyte was expressed by the fluorescence intensity of GFAP, which was also a reflection of the ability of cell facing the stress. There were significant differences between the control group and group SAH in all brain regions undergoing SAH surgery (*p* < 0.01) (Figure 2c–e). The number of neurons activated by SAH were expressed by the fluorescence intensity of C-fos, which was also a reflection of the intensity of cell stress. It could be seen that the fluorescence of all brain regions undergoing SAH surgery is strong, and the difference between SAH and SAH+Thiamet-G group in the EC region was statistically significant (*p* < 0.01) (Figure 3a–c). ATF6, as the key protein of a pathway in the UPR, had a high level of expression in the whole brain. The expression of ATF6 had a significant difference between the SAH group and SAH+Thiamet-G group (*p* < 0.01) (Figure 4a–c).

Xbp1s, as the spliced Xbp1, was not expressed at a high level in the whole brain, but the expression of Xbp1s had a significant difference in between the control group and SAH (*p* < 0.01). The expression of Xbp1s in the EC brain region is different in SAH group and SAH+Thiamet-G group(*p* < 0.01) (Figure 5a,c). The expression level of O-GlcNAc in the whole brain was high. The expression levels of O-GlcNAc were significantly higher in all brain regions of the SAH group compared to the control group (*p* < 0.01). In addition, the expression level of O-GlcNAc in the EC brain region was higher in the SAH+Thiamet-G group compared to the SAH group (*p* < 0.01). (Figure 5b,d).

### 3.3. Xbp1 Reulated O-GlcNAc Modification, and the Up-Regulating of O-GlcNAc Modification Increased the Ability of HT22 Cells to Cope with OGD Treatment

Ctrl-sgRNA group was significantly different from Xbp1-sgRNA under normal treatment (*p* < 0.01). The pLenti-Control-sgRNA did not have a significant effect on HT22 cells. The application of pLenti-CRISPRv2-sgRNA reduced the expression of Xbp1 in HT22 cells. After OGD treatment, there was significant difference in the expression of Xbp1 and GFAT1 between Ctrl-sgRNA group and Xbp1-sgRNA group (*p* < 0.01) (Figure 6a). The expression levels of Xbp1 and GFAT1 were significantly higher in the Xbp1-plasma group compared to the Ctrl-plasmid group under normal treatment. (*p* < 0.05). The expression of Xbp1 and GFAT1 in Xbp1-plasma group was increased, which proved that the overexpression of HT22 cells was successful, and the up-regulation of Xbp1 could increase the expression of the HBP pathway rate-limiting enzyme GFAT1. The expression levels of Xbp1 and GFAT1 were also significantly different between the Ctrl-plasmid group and Xbp1-plasma group under OGD treatment. (*p* < 0.01) (Figure 6b). The immunofluorescence intensity was expressed by the ratio of O-GlcNAc fluorescence intensity in the target area to the control group. The immunofluorescence intensity of the control group was significantly different from that of the Xbp1-sgRNA group and the Xbp1-plasma group (*p* < 0.01). Up-regulation of xbp1 resulted in an increase in the level of O-GlcNAc modification, while down-regulation of xbp1 resulted in a decrease in the level of O-GlcNAc modification (Figure 6c,e). Xbp1-sgRNA group and Xbp1-plasma group had significant differences in the level of apoptosis after OGD treatment (*p* < 0.01). Silencing of Xbp1 led to the increase of apoptosis, while overexpressed of Xbp1 led to the decrease of apoptosis (Figure 6d,f).

## 4. Discussion

We simulated a subarachnoid hemorrhage (SAH) model in mice by inducing an arterial puncture in the brain. Our study aimed to investigate the resulting changes in brain injury, as well as the relationship and role of two factors—the endoplasmic reticulum stress marker Xbp1s and the protein O-GlcNAc modification—in the context of SAH. In this study, red blood cells could be seen in the subarachnoid space in HE staining, which proved that the model of subarachnoid hemorrhage in mice was successful (Figure 1a,b). Based on the SAH models, we found that endoplasmic reticulum stress and O-GlcNAcylation of protein were involved in the nerve injuries and behavioral changes caused by SAH (Figure 1c–e). Then, we used tissue array to observe the expression changes of Xbp1 and O-GlcNAcylation in different brain regions. Tissue array had eliminated the error of each experimental operation between individual samples and presented the expression of brain proteins in each group under a scene. We observed a significant activation of astrocytes and stress neurons in the SAH group, indicating that the SAH process caused severe damage to the mouse nervous system. However, increasing the O-GlcNAc protein modification level in mice using Thiamet-G improved this symptom (Figure 2e and Figure 3c). Similarly, we observed a similarity between the protein immunofluorescence of Xbp1s and O-GlcNAc protein modification in the SAH group, while other related proteins involved in ER stress did not exhibit this characteristic (Figure 5a,b). It is noteworthy that we observed an enhancement of the strength of nerve fibers in the immunofluorescence of ATF6, which may become the direction of our future experiments in investigating whether such signals are transmitted through nerve fibers (Figure 4b). We used cell transfection technology to further explore the relationship between Xbp1 and O-GlcNAcylation (Figure 6a,b). By regulating the expression of Xbp1 in the mouse hippocampal neuron cell line, the effects of Xbp1 and O-GlcNAcylation on HT22 cells were observed. We observed a reduction in O-GlcNAc modification of stressed neuronal cells with up-regulated Xbp1, decreased apoptosis, and increased stress tolerance, while the O-GlcNAc modification increased and stress tolerance increased while apoptosis decreased in mice with silenced Xbp1 (Figure 6c,d).

Subarachnoid hemorrhage (SAH) is the third most common subtype of stroke. Approximately one quarter of patients with SAH die prior to hospital admission; overall, outcomes are improved in those admitted to hospital, but with elevated risk of long-term neuropsychiatric sequelae such as depression [1]. SAH can trigger an inflammatory response and vasospasm in the brain. Blood entering the subarachnoid space damages blood vessels and tissue, releasing pro-inflammatory cytokines. This inflammatory surge contributes to vasospasm, where brain blood vessels constrict, reducing blood flow, and causing ischemia and neuronal damage. Vasospasm is a common complication of SAH and can occur within a few days to several weeks after the initial bleeding event [14,15]. SAH caused neuroinflammation and extensive endoplasmic reticulum stress [2,4]. The binding immunoglobulin protein (Bip) located in the endoplasmic reticulum is a classic marker of endoplasmic reticulum stress in the current study. The level of Bip expression reflects the level of endoplasmic reticulum stress [16,17]. Among the UPR pathways activated by ER stress, the lighter ER stress is regulated by ATF6 and IRE1 pathway, and the stronger ER stress is regulated by PERK pathway [18]. The PERK pathway can lead to the global translation inhibition of cell proteins. When the stress outside the cell continues, it will lead to the occurrence of cell apoptosis [19].

Similar to phosphorylation of protein, O-GlcNAcylation of protein is a protein-modification mode after protein transcription and translation. O-GlcNAcylation can change the position of specific proteins in different cell. Additionally, the proteins modified by O-GlcNAcylation will also show different functions. At the same time, it can also regulate the cell at the transcriptional level by changing the interaction between the modified proteins and other proteins or DNA strands. O-GlcNAc mylation can also change the half-life period of protein, and directly affect the activity of protein in the cell [20,21,22]. The above changes, as a widely dynamic protein modification phenomenon, participate in various cell signal transduction under cell stress, and play a vital role in SAH. Thiamet-G is an inhibitor of O-GlcNAcase. After using Thiamet-G to up-regulate the level of glycosylation modification of protein in tissues, the brain edema of mice was reduced and neurobehavioral changes were improved, indicating that the acute rise of glycosylation modification improved the acute stress ability of nerve cells under SAH.

The O-GlcNAcylation is highly sensitive to the change of intracellular environment. Under the condition of SAH, the level of intracellular O-GlcNAcylation was significantly up-regulated and activated the corresponding cell signal pathway. The researchers linked the O-GlcNAcylation with various pathways of the cell and found that the up-regulated level of O-GlcNAcylation could participate in regulating the change of intracellular environment and replying to injuries. Thus, the ability of cells to face the external environment was enhanced. OGT (O-GlcNAc Transferase) and OGA (O-GlcNAcase) are enzymes involved in the post-translational modification of proteins through the addition and removal of O-GlcNAc (O-linked β-N-acetylglucosamine) moieties, respectively. Studies had shown that the level of O-GlcNAcylation is related to calcium overload, mitochondrial damage, systemic inflammatory reaction, and oxidative stress at the target cells. Up-regulating the O-GlcNAcylation could regulate the Ca2+-balance of myocardial cells [23]. O-GlcNAcylation could reduce the infarct area caused by ischemia-reperfusion and prevent the formation of mPTP [24]. Whether OGT was used to enhance the O-GlcNAcylation or OGA was used to weaken O-GlcNAcylation, the mRNA level of superoxide dismutase or glutathione peroxidase was significantly changed [25,26]. O-GlcNAcylation also participated in the functional stability of regulatory T cells, reduced the number of microglial cells and macrophages in the ischemic brain, and inhibited its LPS-stimulated inflammatory response [27,28].

The substrate material of O-GlcNAcylation mainly comes from HBP, and approximately 5% glucose flux in cells enters HBP. HBP is divided into four key steps. Finally, fructose-6 phosphate is converted into uridine diphosphate phosphorus-N-acetylglucosamine (Udp-GlcNAc). Udp-GlcNAc is used as the substrate material of O-GlcNAc to generate N-glcnac, which is connected to the hydroxy group of serine or threonine (Ser/Thr) by the o-glycosidic bond on the action of O-GlcNAc converting enzyme, and then regulates the function of protein. The rate-limiting enzyme of HBP is glutamine fructose-6-phosphate intermediate transferase (GFAT), which is expressed and plays a role in different tissues and organs of human body in two different forms. It is regulated by feedback inhibition of Udp-GlcNAc at both transcription and translation levels. Some researchers have found that the level of GFAT1 in Caenorhabditis elegans is up-regulated, which improves the protein balance in the worm and prolongs its life span [29]. In nerve cells, Xbp1 cleavage into Xbp1s is required for cellular activity. Silencing Xbp1 reduces GFAT1 expression and O-GlcNAcylation levels. Xbp1-silenced HT22 cells show increased early apoptosis under OGD treatment, while overexpressing Xbp1 decreases it. Up-regulated Xbp1 in mice nervous systems enhance GFAT1 and O-GlcNAcylation, providing neuroprotection.

The hippocampus is crucial for converting short-term memory to long-term memory, with its main input coming from the entorhinal cortex (EC). EC’s structural and functional changes after subarachnoid hemorrhage (SAH) suggest its importance in cognitive function preservation. However, current treatment methods for EC are limited. Future research should identify specific memory damage sites in the EC and develop targeted treatments to better intervene in early-stage SAH. Thiamet-Gs increase of O-GlcNAcylation levels in the SAH model, particularly in the EC region, providing a new direction for clinical management and treatment of SAH.

## 5. Conclusions

Protein needs post-translation processing on the endoplasmic reticulum and Golgi apparatus to express its activity. The function of the endoplasmic reticulum is closely related to protein modification. The combination of special biochemical functional groups and proteins enables proteins to have more functions to meet the needs of cells. Both the use of Thiamet-G and the increase of HBP rate-limiting enzyme GFAT1 caused by the increase of UPR product Xbp1 have improved the stress ability of nerve cells. Xbp1-HBP/O-GlcNAc pathway was activated under the stimulation of SAH, which up-regulated the level of O-GlcNAcylation of cells and enhanced the ability of cells to cope with stress. Xbp1 connected endoplasmic reticulum stress with O-GlcNAcylation of protein at the transcription level and promoted cell survival. The focus of future research is to discover the target key proteins modified by O-GlcNAcylation and efficiently and accurately regulate the modification level of the O-GlcNAcylation of these proteins.

## Figures and Tables

**Figure 1 biomedicines-11-01259-f001:**
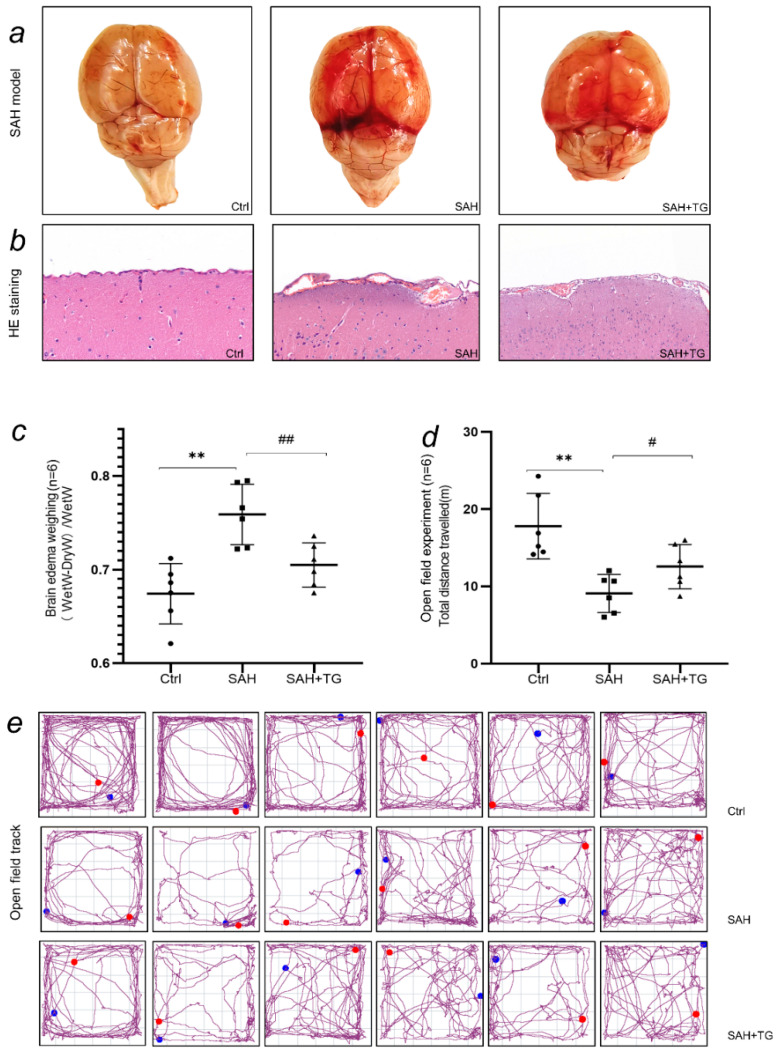
SAH lead to endoplasmic reticulum stress, which induced neuronal injury. Thiamet-G pretreatment can alleviate this nerve injury, and showed the behavioral changes in different groups. (**a**) These brain tissues were not separated from the meninges, and blood residue could be seen in the subarachnoid space of the mouse brain after being treated by the SAH group. (**b**) Verification of SAH model. There was blood retention in the subarachnoid space. The HE staining results showed that there were red blood cells aggregation in the subarachnoid space. Neurons near the cortex after SAH operation was swelling. (**c**) Showed the brain edema weighing after SAH. SAH and SAH+Thiamet-G were subjected to SAH operation. SAH+Thiamet-G was given Thiamet G 1 h before operation. After 12 h reperfusion, the animals were sacrificed. (**d**) Showed the changes in the behavior in mice. The total distance traveled was used to represent the change of mice movement ability. (**e**) The movement track of each mouse was shown in the figure. The mice in the S group exhibited regular movement patterns in the open field experiment, following a consistent path and spending a significant amount of time in specific areas. In contrast, the mice in the SAH group displayed erratic movement, moving unpredictably, and covering a large area without spending much time in any one location. However, the mice in the SAH+Thiamet-G group showed slightly more regular movement patterns than those in the SAH group, suggesting that the treatment may have had some positive effect on their behavior. ^#^
*p* < 0.05, ^##^
*p*, ** *p* < 0.01.

**Figure 2 biomedicines-11-01259-f002:**
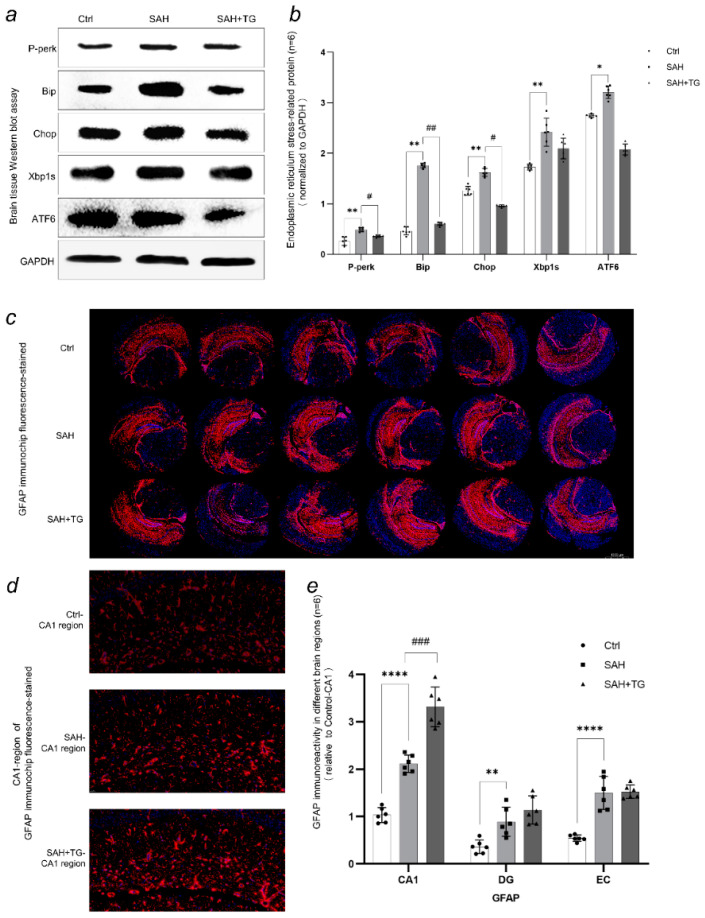
Expression and distribution of protein in mouse brain tissue, and GFAP immunofluorescence chip results. (**a**,**b**) Expression of unfolded protein response related proteins. SAH induced endoplasmic reticulum stress, resulting in the activation of unfolded protein response. Data was expressed as the mean ± SEM. The expression level of the target protein was expressed by the ratio of the gray value of the target protein band to the gray value of the GAPDH band. * *p*, ^#^
*p* < 0.05, ** *p*, ^##^
*p* < 0.01. (**c**) The number and distribution of activated astrocytes was expressed by the immunofluorescence intensity of GFAP staining. GFAP is a protein that is specifically present in astrocytes and can be used as a marker for these cells. Astrocytes are a type of glial cell that plays a crucial role in the central nervous system, and their activation has been linked to various neurological conditions. Therefore, by using GFAP as an indicator, researchers can track astrocyte activation and gain insights into its potential role in disease pathology or other physiological processes [12]. (**d**) Different expression and distribution of GFAP in CA1 region. It can be observed that there was a significant amount of astrocyte activation in the SAH group, while the SAH+Thiamet-G group showed even more widespread astrocyte activation, providing greater protection to the neuronal cells. (**e**) Showed different expression and distribution of GFAP in CA1, DG and EC region. The immunofluorescence intensity of the target region was expressed by the ratio of the fluorescence intensity to the CA1 region of the control group. We specifically chose these three brain regions because during the process of continuous slicing, it was observed that these areas exhibited more significant and meaningful responses compared to other brain regions. After SAH surgery in mice, most brain regions showed astrocyte activation. However, in the CA1 region of the SAH+Thiamet-G group, there was a statistically significant upward trend compared to the SAH group. ^###^
*p* < 0.001, ** *p* < 0.01, **** *p* < 0.0001.

**Figure 3 biomedicines-11-01259-f003:**
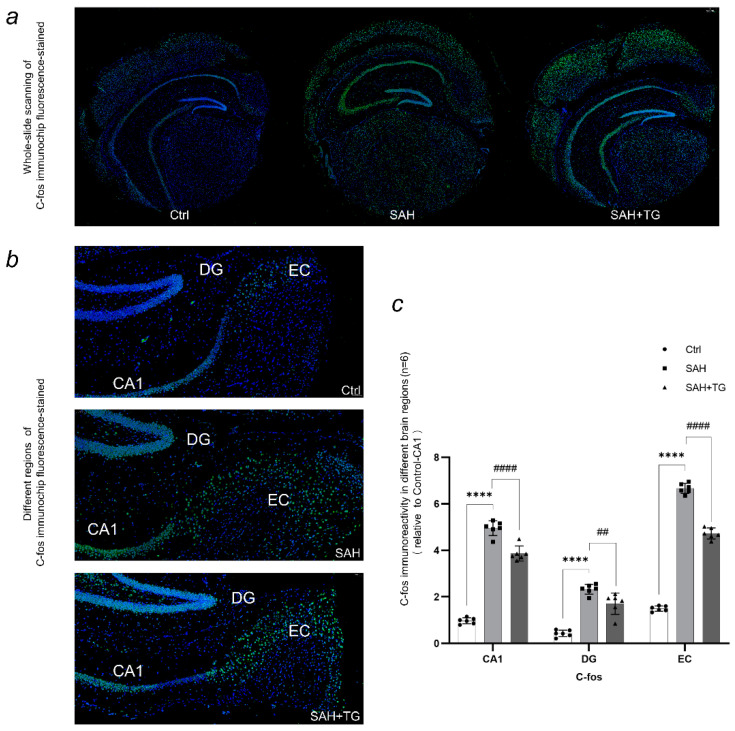
The number and distribution of stressed neurons in mice was expressed by the immunofluorescence intensity of c-fos staining. c-fos is an immediate-early gene that is rapidly expressed in response to various stimuli, including stress. Therefore, C-fos staining can be used as a marker to identify which neurons are activated in response to stress [13]. (**a**) C-fos staining around the hippocampus reveals that SAH surgery activated a large number of neurons in the hippocampus, as well as in the cortex and other brain regions, indicating that most neurons in the mouse brain were under a stress state; (**b**,**c**) The immunofluorescence staining of c-fos in the EC, DG, and CA1 brain regions shows that these three areas, which are related to mouse memory behavior, were specifically activated by SAH, indicating that the neurons in these areas were under a stress state. However, in the SAH+Thiamet-G group, there was a decrease in stress-activated neurons compared to the SAH group, suggesting that the use of Thiamet-G had a protective effect on the neurons. ^##^
*p* < 0.01, ^####^
*p* < 0.0001, **** *p* < 0.0001.

**Figure 4 biomedicines-11-01259-f004:**
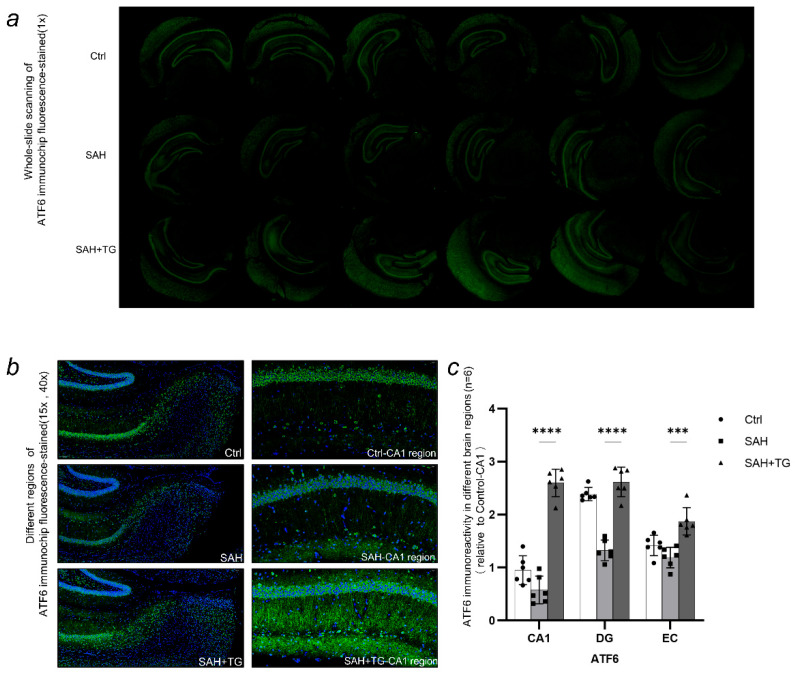
ATF6 (activating transcription factor 6) is a key regulator of the unfolded protein response to endoplasmic reticulum stress. When the endoplasmic reticulum is under stress, unfolded or misfolded proteins accumulate and activate the UPR, which initiates a series of cellular responses to restore ER homeostasis. ATF6 is one of the three major transducers of the UPR and is responsible for regulating the transcription of genes that promote ER protein folding and degradation, as well as genes that enhance ER membrane biogenesis. Through its transcriptional activity, ATF6 plays a crucial role in maintaining ER function and cellular homeostasis under ER stress conditions. (**a**) The global image of the immunofluorescence chip shows that the fluorescence intensity of the SAH group is slightly lower than that of the control group, while the fluorescence intensity of the SAH+Thiamet-G group is higher. (**b**) We observed a specific fluorescence intensity increase in the CA1 region of the SAH+Thiamet-G group, particularly in the nerve fibers, which led us to speculate about the involvement of ATF6. Specifically, we hypothesized that ATF6 could be transferred from regions with lower damage and higher O-GlcNAc modification levels in nerve fibers to regions with higher damage, thereby enhancing protein folding capacity and providing a protective and reparative effect for the neurons. This will be an area of investigation in our future experiments. (**c**) When using Thiamet-G to increase the level of O-GlcNAc modification in mouse brain tissue, the corresponding regions of ATF6 showed an increase compared to the SAH group. *** *p* < 0.001, **** *p* < 0.0001.

**Figure 5 biomedicines-11-01259-f005:**
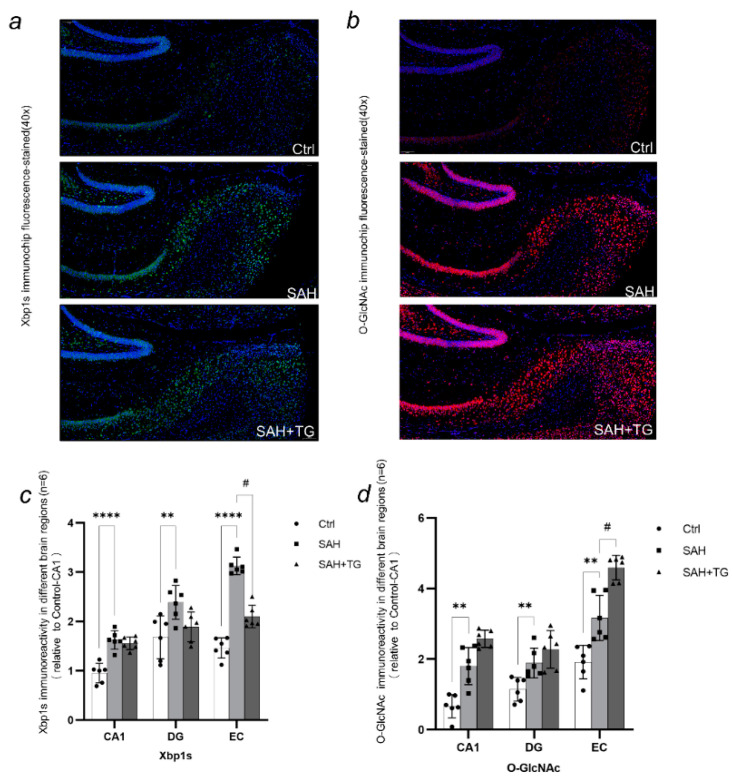
Xbp1s is a transcription factor that plays a crucial role in the unfolded protein response (UPR) signaling pathway of the endoplasmic reticulum (ER). During ER stress, Xbp1 mRNA undergoes an unconventional splicing process to produce the active spliced form Xbp1s, which translocates to the nucleus and regulates the expression of genes involved in ER function and protein folding. (**a**,**c**) In the SAH group, the expression of Xbp1s was increased, reflecting the activation of Xbp1s due to endoplasmic reticulum stress caused by SAH. However, after using the O-glcNAc modification enzyme inhibitor Thiamet-G, the expression of Xbp1s was alleviated, indicating the effect of Thiamet-G in reducing the expression of endoplasmic reticulum stress product Xbp1s. (**b**,**d**) SAH-induced ER stress leads to an increase in protein O-glcNAcylation levels in the SAH group. The use of Thiamet-G also resulted in an increase in O-GlcNAcylation levels in mouse brain tissue. Xbp1s and O-GlcNAcylation showed striking similarities in expression and localization in each brain region after SAH. ^#^
*p* < 0.05, ** *p* < 0.01, **** *p* < 0.01.

**Figure 6 biomedicines-11-01259-f006:**
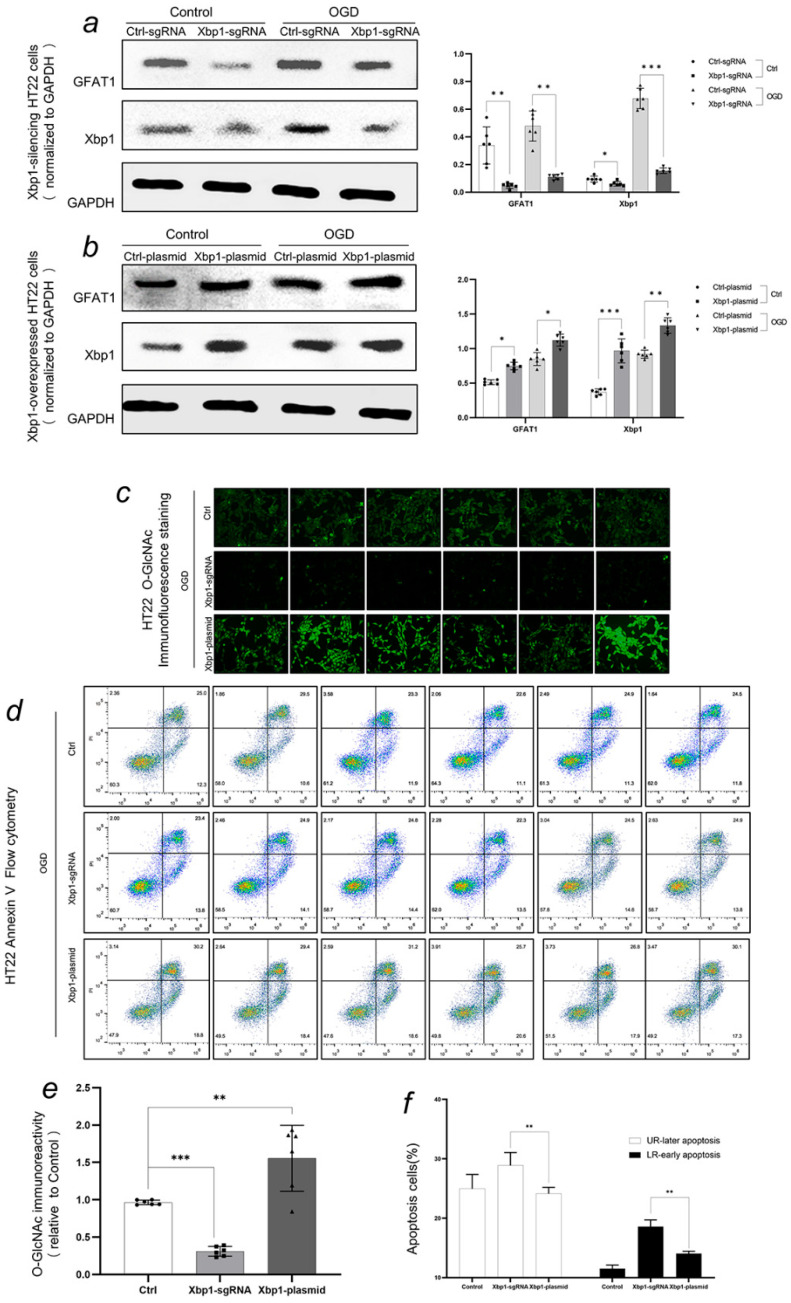
Xbp1s induces GFAT1 expression, a rate-limiting enzyme in the hexosamine pathway, and upregulates O-GlcNAc modification levels, providing neuroprotection. (**a**) HT22 cell line silencing verification showed reduced Xbp1 and GFAT expression. Immunofluorescence intensity was expressed as the target protein to GAPDH gray value ratio. (**b**) HT22 cell line overexpression verification demonstrated increased Xbp1 and GFAT1 expression. Immunofluorescence intensity was similarly represented. (**c**,**e**) O-GlcNAc modification levels in HT22 cells were assessed, with immunofluorescence levels expressed as the target fluorescence level to control group ratio. (**d**,**f**) Phosphatidylserine (PS) is a negatively charged phospholipid in the inner cell membrane surface. Early cell apoptosis changes occur on the membrane surface. Annexin V, a Ca2+-dependent phospholipid-binding protein, binds to PS. Cell membrane integrity and apoptosis can be detected through Annexin V and PS binding. The stress response to OGD treatment was assessed in HT22-Control, HT22-silencing xbp1, and HT22-overexpressing xbp1 groups. The main detection index was the proportion of early and late apoptosis to the total cell count. LR represented the early apoptosis quadrant, while UR represented the late apoptosis and death quadrant. * *p* < 0.1, ** *p* < 0.01, *** *p* < 0.001.

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
